# Substantiation of Methods for Predicting Success in Artistic Swimming

**DOI:** 10.3390/ijerph18168739

**Published:** 2021-08-19

**Authors:** Olha Podrihalo, Leonid Podrigalo, Władysław Jagiełło, Sergii Iermakov, Tetiana Yermakova

**Affiliations:** 1Department of Biological Science, Kharkiv State Academy of Physical Culture, 61022 Kharkiv, Ukraine; rovnayaolga77@ukr.net; 2Department of Medical Science, Kharkiv State Academy of Physical Culture, 61022 Kharkiv, Ukraine; l.podrigalo@mail.ru; 3Department of Sport, Gdansk University of Physical Education and Sports, 80-854 Gdansk, Poland; wjagiello1@wp.pl (W.J.); sportart@gmail.com (S.I.); 4Department of Pedagogy, Kharkiv State Academy of Design and Arts, 61002 Kharkiv, Ukraine

**Keywords:** artistic swimming, prognosis, success, morphofunctional, indicators, indices, functional tests

## Abstract

To develop a methodology for predicting success in artistic swimming based on a set of morphofunctional indicators and indices, 30 schoolgirls, average age (12.00 ± 0.22), were divided into two groups. Group 1: 15 athletes, training experience 4–5 years. Group 2: 15 schoolgirls without training experience. For each participant, we determined the length and weight of the body, the circumference of the chest, vital lung capacity, and the circumference of the biceps in a tense and at rest. The Erisman index, biceps index, and the ratio of proper and actual vital lung capacity was calculated. Them, we conducted the Stange and Genchi hypoxic tests, and flexibility tests for “Split”, “Crab position”, and “Forward bend”. Prediction was conducted using the Wald test with the calculation of predictive coefficients and their informativeness. A predictive table containing results of functional tests and indices of artistic swimming athletes is developed. It includes nine criteria, which informativeness varied in the range of 395.70–31.98. The content of the prediction consists of evaluating the results, determining the appropriate predictive coefficient, and summing these coefficients before reaching one of the predictive thresholds. The conducted research allowed us to substantiate and develop a method for predicting the success of female athletes with the use of morphofunctional indicators and indices.

## 1. Introduction

The selection and prediction of the growth of athletes’ skills are the priority tasks of modern sports science [1,2,3,4]. Predicting, in a broad sense, means anticipatory reflection of the future and the identification of trends in the dynamics of a particular object based on an analysis of its condition in the past and present. The development of the prediction in the narrow sense is a special scientific research of concrete development prospects of any phenomenon. Predicting sports performance involves identifying predictors of success, analyzing their informativeness and making the dependent between them.

Sports prediction is an important part of sports statistics, the development of which is closely related to interdisciplinary integration. Huang and Shen [5] analyzed research on sports prediction and highlighted the main problems and shortcomings. It is concluded that the theory of sports predicting requires improvement.

Another study analyzed the possibility of applying different methods for sports prediction of results [6]. The authors concluded that the combination of methods significantly increases the probability of prediction. The study by Roberts et al. [7] was dedicated to identifying talented athletes based on predicting. The review and meta-analysis revealed a key topic of “coach instinct” as a main component of talent identification decisions.

Other authors suggest that the application of various mathematical methods allows us to obtain more accurate predictions in sports than subjective expert assessments [8]. It is argued that statistics and analytical methods are becoming increasingly important in basketball [9]. It is determined that players’ performance prediction is a serious problem. The authors propose a methodology based on methods of processing rare and irregular data. The results demonstrate the competitiveness of the approach used.

Analyzing the features of sports prediction, Aldous [10] concludes that the use of special algorithms is important. Sports prediction should consider the rating system, answering questions arising in the process of model analysis. Other researchers developed a method for predicting an increase in the efficiency of training 400 m runners [11]. This method allows us to quickly assess the dynamics of physical fitness. It ensures the reliability and quality of prediction based on the training plan.

Other studies present approaches to assess the probability of talent selection in handball [12]. The system used included the general and special tests; assessments made by qualified professionals during training camp; and analysis of athletes’ activities in the game based on video. The results of tests revealed the major coincidence of predictions.

Kalina et al. [13] justified the use of various tests to diagnose the abilities and capabilities of athletes. A conclusion is made about the effectiveness of hardware and simulation techniques. Another study considered predicting the results of an anaerobic sprint and 800 m running test at critical speed [14]. The ability of anaerobic distance was determined to be a significant predictor of anaerobic sprint test results and 800 m running results.

Thus, the available literature suggests the possibility of predicting success in sports based on the use of functional tests and the application of various mathematical methods. However, in artistic swimming (AS), this problem does not yet have a final solution. This determined the relevance of this study.

The aim of the study is to develop a method for predicting the success in artistic swimming based on a set of morphofunctional indicators and indices.

## 2. Materials and Methods

### 2.1. Ethics Statement and Participants 

This study was approved by the Bioethics Committee for Clinical Research and conducted according to the Declaration of Helsinki (protocol of the Commission on Bioethics of the Kharkiv State Academy of Physical Culture No. 25).

The results of the survey of 30 schoolgirls, average age (12.00 ± 0.22) was used as the main materials of the study. The participants were divided into two groups. The first group included 15 AS athletes with 4–5 years training experience, and the second group included 15 schoolgirls who are not engaged in sports. All participants and their parents were informed about the purpose and objectives of the study, informed about the absence of possible harm to their health and gave the written consent to participate in the study. 

### 2.2. The Design of the Study

The following parameters were determined to assess physical development: the body length (BL) and body weight (BW), chest circumference (CC) in the pause, vital lung capacity (VLC), and biceps circumference in a tense and at rest. The measurements were performed according to the requirements of the international unified methodology of anthropometric research [15].

The level and harmony of physical development was determined using the available age and gender standards of physical development of schoolchildren [16]. The regression scale method was used. The interval represented by body length determined the level of physical development. Development was considered harmonious if body weight and chest circumference were in the range M ± δ_R_.

The following indices of physical development were calculated: 

(1) Erisman index
ie = t − 0.5 · l
(1)
where ie is the index, t is the chest circumference at rest (cm), and l is the body length (cm).

(2) The proper vital lung capacity (pvlc) was calculated using the formula:
pvlc = (l · 0.041) − (b · 0.018) − 3.7
(2)
where l is the body length (cm) and b is age (years). 

The ratio of pvlc to actual vlc was found.

(3) biceps index
ib = (lb1 − lb2)/lb2
(3)
where lb1 is the biceps circumference in a tense condition (cm) and lb2 is the biceps circumference at rest (cm).

Hypoxic Stange (inhalation delay time, s) and Genchi (exhalation delay time, s) tests were used to assess the functional condition of the respiratory system.

The “Splits”, “Crab position”, and “Forward bend” tests were used as specific functional tests. These tests are used for selection in the AP according to the current training programs in this sport.

The “Splits” test: The participant is asked to perform the forward split. Behind her is a tripod, on which bars lie on the head. The distance from the floor to the inguinal region (cm) is measured. In the AS athletes, the leg is put forward, the heel resting on the gymnastic bench.

The “Crab position” test: The athlete is lying flat on her back. The athlete pulls her feet close to her buttocks, rests her hands at shoulder level and stretches upwards. The distance between the palms and heels (cm) is measured. 

The “Forward bend” test: The athlete is in a standing position, feet together. The athlete bends forward while holding the leg grip. The bend time (s) is fixed. 

### 2.3. Statistical Analysis

The analysis of the obtained data was performed using licensed MS Excel. The Wald test was used as a tool for solving the predictive problem [17,18]. The method is a predictive table, which includes predictive coefficients of signs and their informativeness. Predictive coefficients were calculated using the formula:
pc = 10 · log {[p1 · (d1/s)]/[p2 · (d2/s)]}
(4)
where pc is a predictive coefficient; s is the total number of people in the group; d1 is the number of test persons who had a value more than the average value in group 1; d2 is the number of test persons who had a value more than the average value in group 2; p1 is the probability of exceeding the average value in group 1; and p2 is the probability of exceeding the average value in group 2.

The predictive coefficients, in the case of the value being lower than average, were found similarly.

The informativeness is calculated by the formula:
i = pc · 0.5 · {[p1 · (d1/s)]/[p2 · (d2/s)]}
(5)
where i is the informativeness, and other designations are the same as in the previous Formula (4).

## 3. Results

The developed methodology for predicting success is given in Table 1.

The predictive table includes the indicator title, the value of its predictive coefficients and the value of informativeness. The indicators in the table are arranged in descending order of informativeness. This minimizes the number of steps in the predicting procedure and reduces the number of possible errors. In the case of the same value of informativeness, the order of location is determined randomly. The value of informativeness below 30.0 is considered insignificant. Indicators with the same or less informativeness are excluded in the table. 

The individual prediction is made by successive summation of the values of the predictive coefficients peculiar to the inspected before reaching one of the thresholds. When the indicator specified in the table is performed, the availability indicator is summed. When the indicator specified in the table is not performed, the absence factor is summed. The value of the allowable error is 5%, which corresponds to the value of the threshold of 13 points. Upon reaching the threshold (+13), a conclusion is made about the high probability of success and growth of athletes’ skills (*p* < 0.05). Upon reaching the threshold (−13), a conclusion is made about the low probability of success (*p* < 0.05). If, after completion of the table, none of the thresholds are reached, it is concluded that the prediction is uncertain and additional research is necessary.

## 4. Discussion

Artistic swimming is a unique sport with complex choreographic exercises performed both above the water surface and underwater. This sport requires athletes to have a high level of physical training, mastery of complex technical skills, and artistry. Viana et al. [19] emphasizes the high physical and physiological requirements for the athlete in this sport. The most significant are the need to adapt to long periods of apnea underwater during performing intense activities.

The development of a methodology for predicting success is most relevant at the stage of preliminary basic training. It is due to the specifics of this stage. Athletes had a certain level of morphofunctional indicator development, important for success. The level of mastering specific technical skills and abilities is still insufficient. Thus, predicting at this stage allows us to determine the feasibility of further training of athletes.

The Wald test is widely used in biomedical research for prediction [17,18]. The application of this method allowed us to develop methods for predicting the success of arm wrestling athletes [20] and kickboxing athletes [21]. The advantages of this method include the ability to choose the probability of prediction [17,18]. The probability of the prediction can vary in the range of 80–99.9%, depending on the selected threshold value (8–30 points). It allows us to significantly increase the efficiency of the prediction. To obtain a reliable prediction, 7–10 indicators in the prediction table are enough. The technique developed by us includes nine indicators. This allows us to consider it informative. 

The sequential analysis procedure requires a comparison of two groups. In the context of our study, these are groups of AS athletes and non-athletes. Such a research design is common in sports science.

Li et al. [22] compared the condition of the elite AS athletes with the condition of non-athlete female students of the same age. A similar design was used in another study [23]. The authors compared anthropometric data of preschool children. It is a positive effect of the AS classes on the physical development of children. Another review presents data from the analysis of physiological parameters of the AS athletes and persons who did not engage in this sport [24]. The authors note the need to standardize the results.

The developed predictive table includes indices and results of functional tests, which determine the success in AS. This approach allows us to provide an integrated assessment of the condition of athletes, to increase the efficiency of the prediction. Similar data are available in the literature. Solana-Tramunt et al. [25] studied the possibility of monitoring the effect of training of AS athletes. The conclusion is made about the need for an integrated approach, with the use of various methods and indicators. 

The available results confirm the legitimacy of including the results of functional tests in the predictive methodology. Escriva-Selles et al. [26] note the prospects of using functional tests to assess the effectiveness of training in the AS.

The index method is used for sports selection and prediction [27,28,29]. Harty et al. [30] used a fat mass index for selection in AS. The values of this index were significantly lower than in strength sport athletes. It is assessed as an illustration of the specifics of the impact of sport on the body of female athletes.

The functional state of the respiratory system should be recognized as a major factor in the success of AS athletes. The index of the ratio of actual and proper VLC and the results of Stange and Genchi tests are used for predicting. An increase in this index confirms the growth of functionality. The results of functional tests exceeding the age norm illustrate high resistance to hypoxia. It reflects the expansion of the functionality of athletes.

The data available in the literature confirm the validity of these assumptions. Garcia et al. [28] reported an increase in hypoxia resistance of the elite AS athletes compared to age standards.

Rovnaya et al. [31] confirmed the increased functionality of the external respiratory system of AS athletes. There was a direct correlation between the increase in functionality and the level of sportsmanship in the AS. The maximum value of the tidal volume and the minimum value of the respiratory rate were established at all stages of the hypoxic test in elite athletes of artistic swimming in comparison with beginners and sub-elite athletes.

The results of the biceps index obtained by us confirm the high level of development of shoulder muscles in AS athletes. It also illustrates the specifics of training in this sport. Exercises for the arm muscles are an important component of training; this quality is necessary to perform complex technical elements in such a sport as AS. 

Similar results were obtained in another study [32]. The authors conclude that it is necessary to perform weightlifting exercises in the training of the AS athletes. Another study confirmed the importance of shoulder muscle balance in the AS athletes [33].

The deficit of body weight in AS athletes is due to the peculiarities of their morphological and nutritional status. Gracilization is one of the predictors of success in this sport. Its main manifestation is a deficit of body weight due to the reduction in the fat component. This coincides with the available literature data. It is determined that the increase in training experience in AS contributes to the increase in disharmony of physical development due to body weight deficit [34]. 

The assumption confirmed by the Erisman index, which is included in the predictive method. This index reflects the harmony of physical development due to the body muscles. The negative value of this indicator confirms the disharmony of development.

Other researchers provided similar data [35]. The authors determine changes in the homeostatic status of the AS athletes. This refers to changes in body weight under the influence of prolonged and intense training. 

Similar results were obtained by Grznar et al. [36]. The authors concluded that the predictor of success in AS is low body weight (lower percentage of fat). In another study, the features of body composition and nutritional status in AS athletes were investigated [37]. 

A review by Robertson and Mountjoy [38] reported a high prevalence of specific energy deficiency syndrome in sports (RED-S) in AS athletes. This syndrome refers to impaired physiological function, including metabolic rate, menstrual function, bone health, immunity, protein synthesis, and cardiovascular health. It leads to psychological consequences that can either precede (due to restrictive dietary habits) or be the result of RED-S. 

The motor tests “Crab position,” “Forward bend”, and “Splits” allow us to estimate the level of flexibility of AS athletes. The high level of flexibility allows athletes to perform difficult technical elements of such a sport as AS. The importance of this quality for success in AS has also been confirmed by a number of studies. Cho et al. [39] determine that synchronous swimmers had an increased range of motion in the joints of the spine and upper and lower extremities compared to swimming athletes. Other authors have studied the diagnostic validity of various functional tests in the AS athletes to assess functional disorders of the upper extremities [40]. The amplitude of movements of the shoulder and elbow joint can be used as tests of the strength of the upper extremities, criteria for the effectiveness of rehabilitation after injury. 

## 5. Conclusions

Our research allowed us to develop a method for predicting success in the AS. The method is based on the Wald test and includes morphological parameters, results of functional tests, and indices based on them. The proposed method is a simple, informative, and objective tool for monitoring and managing the condition of AS athletes. Determining the indicators used is simple and accessible. It allows us to conclude the availability, clarity, and financial feasibility of the prediction.

The developed methodology can be used by coaches when selecting for artistic swimming, as well as a tool for controlling the fitness of athletes. In monitoring the state of artistic swimming athletes, it is necessary to use a set of indicators reflecting physical development (body weight deficit), indices of the state of the respiratory system (ratio of VLC to proper VLC), muscular system (Erisman index and biceps index), results of tests of the state of the respiratory system (Stange test and Genchi test) and flexibility (tests “Crab position”, “Forward bend”, and “Splits”).

## Figures and Tables

**Table 1 ijerph-18-08739-t001:** Methods for predicting the athletes’ success in artistic swimming.

Indicators	Predictive Coefficients		Informativeness
	Availability	Absence	
The ratio of VLC to the PVLC is more than 1	10.8	−6.7	395.70
Biceps index more 6.7%	10.0	−4.5	300.00
Weight deficit of body in relation to age norms	9.5	−3.7	254.46
The time to perform the “Forward bend” test more than 10 s	6.7	−10.8	245.30
Splits less than 0 cm	6.7	−10.8	245.30
The result of the Stange test higher than the age norm, s	3.8	−5.2	88.72
The result of the Genchi test higher than the age norm, s	3.0	−4.8	60.21
The Erisman index less than 0 cm	2.4	−8.5	48.61
The “Crab position” test result is less than 60 cm	1.9	−7.8	31.98

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
