# Peer review of "Substantiation of Methods for Predicting Success in Artistic Swimming"

_ijerph, 2021, doi:10.3390/ijerph18168739_

Round 1
Reviewer 1 Report
The authors developed a method for predicting success in the artistic swimming. However data presented was only based on predictive coefficients and the value of informativeness, not the real success rate. Further experiment data is needed to support the method used.
Author Response
We agree with Distinguished Reviewer 1 that data can be based on real success rates. We show that our approach (as well as that proposed by Reviewer 1) is based on the performance of female athletes who have already achieved success in this sport.
The aim of the study was precisely the development of a predicting methodology in artistic swimming. The calculation of prognostic coefficients is based on a comparison of the morphological and functional indicators of athletes who have already achieved success in this sport, and those who have an almost average level of the indicators used. The proposed methodology allows to make a selection in this sport. Sequential analysis according to Wald allows to determine the likelihood of achieving an event. In our case – this is the probability of achieving or not achieving success. This happens when one of the predictive thresholds is reached. Therefore, the available indicators are sufficient to implement the methodology.
Reviewer 2 Report
The authors ’have undertaken a comprehensive process in the development and evaluation of athletes. It is helpful for researchers and practitioners to be aware of these methods. The topic of the paper can be interesting globally.
Comments:
Would be useful to indicate the aim of the work in the introduction.
It is appropriate that the authors identify which morphological indicators and their parameters are the main for monitoring and managing the condition of the AS athletes in conclusion. Sincerely.
Author Response
Our answer on Comment 1. We have made changes about the aim of our research and it has the following form: The aim of the study is to develop predicting the success method in artistic swimming based on the basis of a set of morphofunctional indicators and indices.
Our answer on Comment 2. We have made changes in section 5. Conclusions. Now it is the second paragraph in this section: In monitoring the state of artistic swimming athletes, it is necessary to use a set of indicators reflecting physical development (body weight deficit), indices of the state of the respiratory system (ratio of VLC to proper VLC), muscular system (Erisman index, biceps index), results of tests of the state of the respiratory system (Stange test and Genchi test) and flexibility (tests "Crab position", "Forward bend", "Splits").
Reviewer 3 Report
Abstract. It is recommended to expand the results section. Also be more specific with the conclusions of the study
Introduction. Is correct.
Materials and Methods. Line 102- to 110 and 132 to 145: It is necessary to improve the format of the formulas. This aspect seems to me the weakest in the manuscript.
Results. The result and tables are correct.
Discussion. Line 221-223; 337, 339- Supplement these statements with the data from the studies cited.
Conclusions: The conclusions must be expanded and specified. Would it be possible to add a real practical application for coaches?
References. Check or changed the references: 5, 16, 18, 36
Making the indicated modifications, the study is novel and interesting.
Abstract. It is recommended to expand the results section. Also be more specific with the conclusions of the study
Introduction. Is correct.
Materials and Methods. Line 102- to 110 and 132 to 145: It is necessary to improve the format of the formulas. This aspect seems to me the weakest in the manuscript.
Results. The result and tables are correct.
Discussion. Line 221-223; 337, 339- Supplement these statements with the data from the studies cited.
Conclusions: The conclusions must be expanded and specified. Would it be possible to add a real practical application for coaches?
References. Check or changed the references: 5, 16, 18, 36
Making the indicated modifications, the study is novel and interesting.
Author Response
We have reread the manuscript once more and make corrections in the spelling /writing.
Our answer on Comment 1. We agree with Distinguished Reviewer 3 on comment 1. But, there is limitations of the size of the annotation according to the Instruction for authors (A single paragraph of about 200 words maximum).
Our answer on Comment 3. We have made changes in the format of the formulas 1, 2, 3, 4, 5 according to the information taken from MDPI_template.zip - contains the template, logo, class and bibliography style files
(a = b + c + d + e + f + g + h + i + j + k + l + m + n + o + p + q + r + s + t + u + v + w + x + y + z (2)
the text following an equation need not be a new paragraph.)
Our answer on Comment 5. We have added the following text in section 4. Discussion in a paragraph started with the words “Rovnaya et al. [31] confirmed …” - The maximum value of the tidal volume and the minimum value of the respiratory rate were established at all stages of the hypoxic test in elite athletes of artistic swimming in comparison with beginners and sub-elite athletes.
337, 339 – it is number of Reference
Our answer on Comment 6. We add the following text in section 5. Conclusions (it is the second paragraph): The developed methodology can be used by coaches when selecting for artistic swimming, as well as a tool for controlling the fitness of athletes.
Our answer on Comment 7. Reference 5 – we have corrected the text on reference 5 (lines 39-43) and it has the following view: “Huang and Shen [5] analyzed research on sports prediction and highlighted the main problems and shortcomings. It is concluded that the theory of sports predicting requires improvement.”
Reference 16 – we check the content of the text on reference 16. It is a reference on National standards of schoolchildren physical development.
Reference 18 – we check the content of the text on reference 18. It is a reference on monograph which describes the method of sequential analysis according to Wald.
Reference 36 – the paragraph “The assumption confirmed by the Erisman index, which is included in the predictive method. This index reflects the harmony of physical development due to the body muscles. The negative value of this indicator confirms the disharmony of development.” We moved above the paragraph “Other researchers provided similar data [35]. The authors determine changes in the homeostatic status of the AS athletes. It refers to changes in body weight under the influence of prolonged and intense training.” Thus, the context of the text will be logical for understanding.